# Clinical and Molecular Characteristics of *KRAS* Codon-Specific Mutations in Advanced Pancreatic Ductal Adenocarcinoma with Prognostic and Therapeutic Implications

**DOI:** 10.3390/ijms262210908

**Published:** 2025-11-11

**Authors:** Dongwoo Cho, Kabsoo Shin, Tae Ho Hong, Sung Hak Lee, Younghoon Kim, In-Ho Kim, Sook-Hee Hong, MyungAh Lee, Se Jun Park

**Affiliations:** 1Department of Internal Medicine, Seoul St. Mary’s Hospital, College of Medicine, The Catholic University of Korea, Seoul 06591, Republic of Korea; dwcho725@gmail.com; 2Division of Medical Oncology, Department of Internal Medicine, Seoul St. Mary’s Hospital, College of Medicine, The Catholic University of Korea, Seoul 06591, Republic of Korea; agx002@naver.com (K.S.); ihkmd@icloud.com (I.-H.K.); ssuki76@catholic.ac.kr (S.-H.H.); angelamd@catholic.ac.kr (M.L.); 3Cancer Research Institute, College of Medicine, The Catholic University of Korea, Seoul 06591, Republic of Korea; 4Department of General Surgery, Seoul St. Mary’s Hospital, College of Medicine, The Catholic University of Korea, Seoul 06591, Republic of Korea; gshth@catholic.ac.kr; 5Department of Pathology, College of Medicine, The Catholic University of Korea, Seoul St. Mary’s Hospital, Seoul 06591, Republic of Korea; hakjjang@catholic.ac.kr (S.H.L.); younghoonkim@catholic.ac.kr (Y.K.)

**Keywords:** pancreatic cancer, *KRAS* codon-specific mutations, survival outcomes, precision medicine

## Abstract

*KRAS* mutations occur in over 90% of pancreatic ductal adenocarcinomas (PDACs), most commonly at codon 12, but the clinical implications of codon-specific subtypes in advanced disease remain unclear. We retrospectively analyzed 269 patients with advanced PDAC who underwent next-generation sequencing between 2020 and 2024 at a single tertiary cancer center. Clinical features, co-mutations, treatment outcomes, and survival were evaluated. *KRAS* mutations were detected in 92% of patients, most frequently G12V (36%) and G12D (34%), followed by G12R (10%) and non-G12 variants (9%). *TP53* was the most frequent co-mutation (67%) and was significantly enriched in G12D tumors compared with wild type (74.2% vs. 31.8%). Mutations in homologous recombination and DNA damage response genes were more common in non-G12 and wild-type tumors, although not statistically significant. Serum CA 19-9 was elevated in most G12-mutant tumors, whereas approximately 40% of non-G12 and wild-type patients showed normal levels despite advanced disease. No significant survival differences were observed among *KRAS* subtypes in the overall or treated cohorts. However, patients with G12V mutations achieved significantly longer survival with fluorouracil-based than gemcitabine-based chemotherapy. These findings suggest that, while *KRAS* subtyping alone lacks prognostic value, the G12V subtype may inform chemotherapy selection and warrants further prospective validation.

## 1. Introduction

Pancreatic ductal adenocarcinoma (PDAC) is currently the third leading cause of cancer-related mortality and is projected to rise to the second leading cause by 2030 [1,2]. Most patients present with advanced-stage disease, with a median survival of less than one year [3]. Two phase III randomized trials have demonstrated that FOLFIRINOX and gemcitabine plus nab-paclitaxel are superior to gemcitabine monotherapy, establishing them as standard first-line therapies for advanced PDAC [4,5]. Although selecting the optimal regimen can improve survival outcomes, treatment decisions are primarily based on clinical factors such as performance status, comorbidities, and patient preference, as no validated biomarkers are available to guide personalized therapy.

Among the genetic alterations identified in PDAC, oncogenic *KRAS* mutations are the most prevalent, occurring in more than 90% of cases and frequently coexisting with alterations in *TP53*, *CDKN2A*, and *SMAD4* [6,7]. These mutations drive oncogenic cellular processes and influence prognosis, with the KRAS oncoprotein playing a central role in signaling by linking growth factor receptors to intracellular pathways [8]. The majority of *KRAS* missense mutations occur at codon 12, with p.G12D being the most common (35–40%), followed by p.G12V (20–30%), p.G12R (10–20%), and other rare variants [9,10] (subsequently abbreviated as G12D, G12V, and G12R).

Although *KRAS* was historically considered an undruggable target, recent advancements have led to the development of small-molecule inhibitors targeting mutant *KRAS* [11]. Sotorasib and adagrasib, the most clinically advanced *KRAS* G12C inhibitors, have received FDA approval for the treatment of *KRAS* G12C-mutated lung cancer [12,13]. However, *KRAS* G12C mutations are rare in PDAC, occurring in less than 2% of cases, thereby limiting the therapeutic relevance of these agents in pancreatic cancer. In contrast, inhibitors targeting the more prevalent *KRAS* G12D mutation have demonstrated promising tumor regression in preclinical studies and are currently being evaluated in clinical trials [14,15]. Consequently, codon-specific *KRAS* mutations are emerging as predictive biomarkers to guide treatment decisions.

Differences in *KRAS* allele-specific mutations may influence distinct signaling pathways and lead to variations in gene expression during PDAC carcinogenesis. Although each *KRAS* allele subtype exhibits unique biochemical and clinicopathological features, differences among *KRAS* allele subtypes in PDAC have not been well studied [16,17]. While evidence on the prognostic impact of *KRAS*-mutated PDAC subtypes remains limited, recent studies suggest that the *KRAS* G12 subtype is associated with poorer outcomes than *KRAS* non-G12 variants [18], with *KRAS* G12D linked to a worse prognosis than the G12R subtype [10]. However, prior analyses combined heterogeneous disease stages and did not assess relationships with systemic treatment response, leaving the clinical relevance of *KRAS* subtypes uncertain. To properly evaluate codon-specific prognosis, studies in advanced PDAC should determine whether *KRAS* mutation subtypes are associated with survival outcomes and the relative effectiveness of systemic chemotherapy.

In this study, we characterized the prevalence of *KRAS* mutation subtypes and their co-mutations in advanced PDAC and evaluated their associations with clinicopathologic features and prognostic outcomes. Additionally, we further investigated the impact of *KRAS* mutation subtypes on systemic chemotherapy effectiveness and survival outcomes.

## 2. Results

### 2.1. Patients

A total of 269 patients with pathologically confirmed advanced pancreatic cancer who underwent next-generation sequencing (NGS) between June 2020 and December 2024 at Seoul St. Mary’s Hospital were included in this retrospective study (Figure 1).

Among them, 245 patients received systemic chemotherapy and underwent radiological response evaluation, comprising the treated cohort. Baseline characteristics of the entire cohort are summarized in Table 1. The median age was 66 years (IQR, 61–73), and 52.4% of patients were male. A total of 64.3% had initially advanced disease, while 35.7% had recurrent disease, and most patients (82.2%) presented with metastatic disease. The primary tumor was located in the pancreatic head in 50.2% and in the body or tail in 49.8% of patients, and the majority of tumors (89.2%) were histologically classified as ductal adenocarcinoma. Regarding the specimen source for NGS, 65.4% were obtained from primary tumor tissues, 28.3% from metastatic sites, and 6.3% from liquid biopsies, with specimens acquired by surgical resection (42.8%), needle biopsy (50.9%), or peripheral blood (6.3%). *KRAS* mutations were identified in 91.8% of patients, while *TP53*, *CDKN2A*, and *SMAD4* mutations were detected in 64.3%, 29.4%, and 12.3% of cases, respectively. Among the 22 patients with *KRAS* wild-type tumors, actionable genomic alterations are summarized in Appendix A, with druggable alterations such as *BRAF* V600E detected in three patients.

### 2.2. KRAS Mutation Subtypes and Survival Outcomes

Among the 269 patients, 247 (92%) harbored *KRAS* mutations. The most prevalent subtype was G12V (96 patients, 36%), followed by G12D (93 patients, 34%), G12R (28 patients, 10%), and non-G12 codon mutations (23 patients, 9%) (Figure 2A). Two patients had co-occurring G12D and non-G12 mutations. G12C mutations were identified in 6 patients, representing 2% of the total cohort. The median follow-up duration for the entire cohort was 9.9 months (95% CI, 8.9–11.9), and 218 patients (81.0%) had died at the time of analysis. The median overall survival (OS) for the entire cohort was 12.5 months (95% CI, 9.9–13.8). OS according to *KRAS* mutation subtypes revealed no statistically significant difference across groups (*p* = 0.64, Figure 2C). The median OS was 11.2 months (95% CI, 8.6–24.6) in patients with wild-type *KRAS*, 12.5 months (95% CI, 9.2–15.0) in those with G12D mutations, 12.7 months (95% CI, 8.7–14.8) in G12V, 10.1 months (95% CI, 7.8–16.4) in G12R, and 15.4 months (95% CI, 8.9–25.9) in non-G12 mutations. The G12C subgroup showed a numerically longer median OS of 25.9 months (95% CI, 13.8–not reached), though the sample size was small and the difference was not statistically significant. Compared with patients carrying wild-type *KRAS*, none of the specific *KRAS* subtypes showed a statistically significant difference in OS (Figure 2D).

### 2.3. KRAS Allele Specific Mutations and Co-Mutations

To characterize the patterns of co-occurring genomic alterations across different *KRAS* alleles, we generated an oncoplot and a heatmap stratified by *KRAS* mutation subtypes (Figure 3). *TP53*, *CDKN2A*, and *SMAD4* were the most frequently co-altered genes, regardless of *KRAS* subtype. The heatmap displays the frequency of mutations in *TP53*, *CDKN2A*, *SMAD4*, *ARID1A*, and genes associated with homologous recombination deficiency (HRD) and DNA damage response (DDR) pathways across *KRAS* subgroups (Figure 3B). *TP53* was the most commonly altered gene overall, with particularly high frequencies in the G12D and G12R subgroups (74.2% and 67.9%, respectively), whereas its mutation rate was substantially lower in the wild-type group (31.8%). Among all evaluated genes, only *TP53* showed a statistically significant difference in mutation frequency across *KRAS* subtypes (*p* = 0.0074), whereas other genes showed no significant variation, although mutations in HRD- or DDR-related genes tended to be more frequent in the non-G12 and wild-type subgroups. To evaluate subgroup-specific differences in *TP53* mutation frequency, pairwise comparisons demonstrated a significantly higher frequency in the G12D subgroup compared to wild-type *KRAS* (adjusted *p* = 0.0089), whereas no other comparisons showed statistically significant differences.

### 2.4. KRAS Mutation Subtypes and Clinical Characteristics

Baseline clinical characteristics were compared across *KRAS* mutation subtypes, including G12D, G12V, G12R, non-G12 mutations, and wild type (Appendix A). Most characteristics were evenly distributed among the groups, with no statistically significant differences observed. However, serum carbohydrate antigen 19-9 (CA 19-9) levels at the time of advanced disease diagnosis differed significantly according to *KRAS* mutation subtype (*p* = 0.011). A higher proportion of patients with G12D, G12V, or G12R mutations exhibited elevated CA 19-9 levels (77.8%, 85.4%, and 85.2%, respectively), whereas a greater proportion of patients with non-G12 mutations or wild-type *KRAS* showed normal CA 19-9 levels (42.1% and 40.9%, respectively). Although not statistically significant, several trends of interest were noted. Tumors in the wild-type group were more frequently located in the pancreatic head (63.6%, *p* = 0.459), while tumors in the G12R group were more often found in the body or tail of the pancreas (60.7%). Elevated carcinoembryonic antigen (CEA) levels were more frequently observed in the non-G12 group (63.2%, *p* = 0.395), and the neutrophil-to-lymphocyte ratio (NLR) similarly showed no significant differences across the subgroups (*p* = 0.201).

### 2.5. KRAS Mutation Subtypes and Treatment Outcomes

A total of 245 patients were included in the treated cohort, and their baseline characteristics are summarized in Table 2. The majority of patients (69.4%) had 0–1 metastatic organ site, while 30.6% had two or more. The liver was the most frequent site of metastasis (53.4%), followed by the peritoneum (24.9%) and lungs (16.3%). Elevated baseline CA 19-9 levels were observed in approximately three-quarters of patients (71.8%). As first-line palliative chemotherapy, gemcitabine-based regimens were administered to 56.3% of patients, the majority of whom received gemcitabine plus nab-paclitaxel (53.8%), while fluorouracil-based regimens were given to 43.7%, including FOLFIRINOX in 38.7%.

In the overall treated cohort, the median progression-free survival (PFS) was 6.3 months (95% CI, 5.5–7.3), and the median OS was 11.2 months (95% CI, 9.6–13.1). Survival analysis revealed no statistically significant difference in PFS among the *KRAS* mutation subtypes (*p* = 0.93; Figure 4A). Similarly, no significant difference in OS was observed between the subgroups (*p* = 0.37; Figure 4B). Findings from very small subgroups are exploratory and should be interpreted with caution. Compared with *KRAS* wild-type, none of the codon-specific *KRAS* subtypes exhibited statistically significant differences in PFS or OS (Appendix A). Treatment outcomes across *KRAS* mutation subtypes are summarized in Table 3. In the overall cohort, the objective response rate (ORR) was 35.5%, with the highest rate observed in the non-G12 group (55.0%), followed by the G12D (38.8%), G12V (34.1%), and G12R (26.9%) subgroups, and the lowest in the wild-type group (20.0%). Compared with the G12D subgroup, no statistically significant differences in ORR were found for any other subtype. The disease control rate (DCR) was 85.3% in the overall cohort, with no significant differences relative to the G12D subgroup. Although numerically lower DCRs were noted in the G12R (73.1%) and wild-type (75.0%) groups, these differences were not statistically significant. The 6-month PFS rate and 12-month OS rate in the total cohort were 51.8% (95% CI, 45.7–58.6) and 48.0% (95% CI, 41.9–54.8), respectively, with no significant differences in these time-specific survival rates when comparing the G12D subgroup with other mutation subtypes.

### 2.6. Effectiveness of Treatment Regimens

In the overall treated cohort, patients receiving fluorouracil-based chemotherapy had a numerically longer median PFS of 7.8 months (95% CI, 5.7–9.6) compared with 5.8 months (95% CI, 5.3–7.2) for those receiving gemcitabine-based regimens, although the difference was not statistically significant (*p* = 0.12; Figure 5A). Median OS was 13.1 months (95% CI, 10.8–16.0) with fluorouracil-based therapy versus 9.7 months (95% CI, 8.0–12.6) with gemcitabine-based therapy (*p* = 0.08; Figure 5B). No significant differences were observed in ORR (32.7% vs. 37.7%, *p* = 0.50) or DCR (86.0% vs. 84.8%, *p* = 0.86) between the two regimens (Appendix A). In the G12D subgroup, survival outcomes were similar between regimens. Median PFS was 6.9 months (95% CI, 4.8–11.8) with fluorouracil-based therapy and 5.8 months (95% CI, 5.4–7.5) with gemcitabine-based therapy (*p* = 0.45; Figure 5C), while median OS was 13.1 months (95% CI, 8.8–19.6) and 10.5 months (95% CI, 8.3–17.2), respectively (*p* = 0.93; Figure 5D). In the G12V subgroup, patients treated with gemcitabine-based regimens had a significantly inferior median PFS of 5.3 months (95% CI, 3.7–7.6) compared with 8.3 months (95% CI, 6.9–12.8) for those receiving fluorouracil-based regimens (HR = 1.78; 95% CI, 1.13–2.81; *p* = 0.01; Figure 5E). Median OS was also shorter with gemcitabine-based therapy (6.6 months [95% CI, 4.9–13.0]) than with fluorouracil-based therapy (14.8 months [95% CI, 13.0–19.3]) (HR = 1.91; 95% CI, 1.21–3.02; *p* = 0.01; Figure 5F). Although ORR (38.5% vs. 30.6%) and DCR (92.3% vs. 79.6%) were numerically higher with fluorouracil-based therapy, these differences were not statistically significant.

## 3. Discussion

In this retrospective study of patients with advanced PDAC, we comprehensively characterized the distribution of *KRAS* mutation subtypes, their co-mutation profiles, and associated clinical implications, including response to chemotherapy and survival outcomes. Unlike previous studies that assessed the prognostic implications of *KRAS* mutation subtypes but included surgically resected cases, our analysis focused exclusively on patients with advanced disease, thereby reflecting the majority of real-world clinical presentations. No statistically significant differences in OS or PFS were observed across *KRAS* subtypes, both in the overall cohort and within the treated cohort. However, subgroup analysis revealed that patients harboring *KRAS* G12V mutations derived significantly greater survival benefit from fluorouracil-based chemotherapy compared to gemcitabine-based regimens, suggesting a potential subtype-specific therapeutic interaction.

Consistent with prior reports, *KRAS* mutations were observed in over 90% of cases, with G12D and G12V comprising the most frequent subtypes [9,10]. The *KRAS* G12D subtype has historically been associated with more aggressive disease biology and poorer prognosis [10,19]. However, in our study, no statistically significant differences in survival outcomes or treatment effectiveness were observed among the *KRAS* subtypes. These findings align with recent studies reporting no significant OS differences among codon 12 *KRAS* mutants in patients with advanced-stage PDAC [18]. Although previous studies have suggested that the G12R subtype is associated with better survival compared to G12D in advanced disease [20], our analysis did not reveal any significant differences in OS, PFS, or treatment response between the two subtypes. Further validation in larger cohorts with multivariable adjustment for additional prognostic factors is needed to clarify the clinical implications of the G12R subtype. While limited by the very small sample size, the G12C subgroup demonstrated numerically longer OS, which may reflect the inclusion of patients who received *KRAS* G12C-targeted therapies.

Analysis of co-occurring genomic alterations revealed that *TP53* was the most frequently co-mutated gene, present in 67% of *KRAS*-mutated cases. The G12D subtype, in particular, exhibited a significantly higher frequency of *TP53* mutations (74.2%) compared with the *KRAS* wild-type group (31.8%), a finding consistent with previous reports [10]. Co-mutation of *KRAS* and *TP53* has been implicated in promoting PDAC progression and metastasis, suggesting a potential molecular synergy that may underlie the poorer OS associated with the G12D subtype [21]. Alterations in HRD- and DDR-related genes were observed more often in non-G12 and *KRAS* wild-type tumors than in G12-mutant tumors, although this difference was not statistically significant. These findings indicate that distinct *KRAS* alleles may influence the broader genomic landscape and warrant further investigation, particularly regarding the role of HRD and DDR mutations as biomarkers for platinum-based therapy.

No significant differences in clinical characteristics, including histologic differentiation, were observed across *KRAS* subtypes. Serum CA 19-9 levels were significantly elevated in patients with G12 mutations, whereas a substantial proportion of patients with non-G12 or wild-type tumors (~40%) exhibited normal levels at the time of advanced disease diagnosis. This finding, which may relate to differences in *KRAS*-driven oncogenesis and tumor marker production, suggests that in non-G12 or wild-type cases, CA 19-9 may not reliably reflect treatment response.

In the treated cohort, no significant differences in PFS or OS were observed across *KRAS* subtypes, and ORR was slightly higher in the G12D subtype compared with the G12R and wild-type groups, suggesting that the prognostic significance of *KRAS* subtypes in advanced PDAC remains controversial. By treatment regimen, no difference in efficacy was observed between fluorouracil- and gemcitabine-based therapies in the G12D group, whereas patients with the G12V subtype showed significantly improved PFS and OS with fluorouracil-based therapy. These findings suggest that the G12V subtype may be less sensitive to gemcitabine and highlight the importance of investigating associations between *KRAS* mutation subtypes and transcriptome-based predictive signatures, such as GemPred, which predicts benefit from gemcitabine treatment [22]. While these observations require prospective validation, *KRAS* G12V mutation status may serve as a predictive biomarker for fluorouracil-based first-line chemotherapy in pancreatic cancer.

Our study has several limitations. First, its retrospective design and single-institution setting may introduce ascertainment bias and limit the generalizability of the findings. Second, the sample size of certain *KRAS* subgroups, such as G12C and non-G12 mutations, was small, reducing the statistical power to detect meaningful differences. Taken together, these two issues render the findings hypothesis-generating and warrant confirmation in adequately powered, prospective multicenter cohorts. Third, potential confounders such as performance status, disease burden, treatment dose intensity, and subsequent therapies may have influenced treatment outcomes, but multivariable analysis could not be performed owing to the limited sample size. Finally, tumor heterogeneity and temporal evolution of mutational profiles may not have been fully captured by single-sample NGS, and in a subset of patients, liquid biopsy was used, which may further affect consistency of genomic profiling.

Despite these limitations, our study provides comprehensive insights into the clinical and molecular characteristics of *KRAS* codon-specific mutations in advanced PDAC and their relationship with treatment outcomes. Prognostic differences across *KRAS* subtypes were not observed in our cohort, and although a differential response to chemotherapy was noted in the G12V subgroup, the biological rationale remains insufficient and requires further validation. Therefore, *KRAS* subtyping may not yet be essential for therapeutic decision-making in clinical practice. Nevertheless, identifying *KRAS* G12C mutations is important given the availability of effective targeted therapies [23,24], and genomic profiling of *KRAS* wild-type tumors remains essential to detect actionable alterations [25]. Furthermore, as novel *KRAS*-targeted therapies, including G12D-selective inhibitors, enter clinical trials [14,15], defining biologically and clinically meaningful subgroups may become increasingly important.

## 4. Materials and Methods

### 4.1. Patients

Patients with histologically or cytologically confirmed pancreatic exocrine cancer were eligible for inclusion in this study if they had NGS results obtained at Seoul St. Mary’s Hospital. The study was conducted in accordance with Korean regulations and the principles of the Declaration of Helsinki. The Institutional Review Board (IRB) of The Catholic University of Korea, Seoul St. Mary’s Hospital, approved data acquisition (protocol code: KC24RASI0307) and granted a waiver of informed consent due to the retrospective nature of the analysis. Retrospective analysis was conducted on patient demographics, clinical characteristics, pathological findings, genomic profiles, treatment history, and outcomes using data extracted from electronic medical records.

### 4.2. Molecular Alteration Analyses

Molecular profiling and *KRAS* status were determined through tumor tissue analysis using the Oncomine Comprehensive Assay Plus or liquid biopsy via AlphaLiquid100. These panels included 517 and 118 genes, respectively, covering oncogenes, tumor suppressor genes, gene rearrangements, microsatellite stability status, and tumor mutational burden. Formalin-fixed, paraffin-embedded (FFPE) tumor specimens or plasma samples were collected for NGS either at the time of diagnosis or upon disease recurrence. For patient-level analyses (survival and clinicopathologic comparisons), if more than one *KRAS* alteration was detected in a single patient, we assigned a single codon category based on the highest variant allele frequency to avoid double-counting.

Genomic DNA and RNA were extracted from FFPE tumor samples and analyzed using the Oncomine Comprehensive Assay Plus (Thermo Fisher Scientific, Waltham, MA, USA) on the IonTorrent™ S5 XL platform, following the manufacturer’s protocols. Sequencing quality was assessed using plug-in coverage analysis, and data were processed through dedicated bioinformatic workflows within the Ion Reporter v5.12 server (IonTorrent, Waltham, MA, USA). Variants, including single nucleotide variants (SNVs), insertions/deletions (indels), copy number variations (CNVs), and gene fusions, were filtered based on the following thresholds: a minimum allele frequency of ≥1.5% for hotspot SNVs and indels, a minimum read count of 100 for fusions, and CNV thresholds of ≥6 for gains and ≤0.7 for losses. Variants were classified into clinical relevance tiers according to established guidelines [26].

Cell-free DNA (cfDNA) was extracted from plasma samples prior to systemic treatment. DNA NGS library preparation and solution-based target enrichment were performed at IMBdx, Inc. (Seoul, Republic of Korea) using AlphaLiquid100, as previously described [27]. A targeted panel of 118 cancer-related genes was used to detect four variant classes (SNVs, indels, CNVs, and fusions). Captured DNA libraries were sequenced on the Illumina NovaSeq 6000 platform (Illumina, San Diego, CA, USA) in 2 × 150 bp paired-end mode. For AlphaLiquid100, variants were called using thresholds of at least 0.1% allele fraction for SNVs and indels (high-confidence at 0.5% or higher), 2.4 copies or higher for CNVs, and 0.2% or higher for fusions.

### 4.3. Mutational Status of HRD- and DDR-Related Genes

To investigate the association between *KRAS* mutation subtypes and genomic alterations related to HRD and DDR pathways, we evaluated the prevalence of HRD-related, core HRD, and DDR-associated gene mutations across each *KRAS* mutation subgroup. Genomic alterations associated with HRD and DDR pathways were identified according to a curated list of genes implicated in these repair mechanisms. HRD-related alterations were defined as pathogenic variants in genes functionally involved in homologous recombination repair, including *ATM*, *BAP1*, *BARD1*, *BLM*, *BRCA1*, *BRCA2*, *BRIP1*, *CHEK2*, *FAM175A*, *FANCA*, *FANCC*, *NBN*, *PALB2*, *RAD50*, *RAD51*, *RAD51C*, and *RTEL1* [28]. Among these, *BRCA1*, *BRCA2*, and *PALB2* were designated as core HRD genes, reflecting their pivotal roles in double-strand break repair and their clinical relevance as biomarkers for predicting response to platinum and PARP inhibitor therapy. In addition, pathogenic variants in genes involved in homologous recombination and DDR pathways, including *ARID1A*, *ATM*, *ATRX*, *BAP1*, *BARD1*, *BLM*, *BRCA1*, *BRCA2*, *BRIP1*, *CHEK1*, *CHEK2*, *FANCA*, *FANCC*, *FANCD2*, *FANCE*, *FANCF*, *FANCG*, *FANCL*, *MRE11A*, *NBN*, *PALB2*, *RAD50*, *RAD51*, *RAD51B*, and *WRN*, were evaluated as HRD and DDR-associated alterations [29].

### 4.4. Assessments

Tumor assessments were performed in accordance with the Response Evaluation Criteria in Solid Tumors (RECIST) version 1.1. Imaging evaluations were conducted using computed tomography (CT) of the thorax, abdomen, and pelvis at 6- to 8-week intervals following the initiation of chemotherapy. Additional imaging was performed as clinically indicated. Radiologic response-evaluable patients were defined as those who received at least one cycle of chemotherapy and underwent at least one post-baseline disease assessment using imaging modalities. Serum tumor markers, including CA 19-9 and CEA, were measured concurrently with radiologic assessments. For stratified analyses, we applied a CA 19-9 threshold of 59× the upper limit of normal consistent with landmark clinical trials [4,5].

### 4.5. Statistical Analysis

Descriptive statistics are reported as proportions or medians with interquartile ranges (IQR). Categorical variables were compared using the chi-square test or Fisher’s exact test, while continuous variables were analyzed using Student’s *t*-test. For pairwise comparisons of categorical variables, Fisher’s exact test was employed with false discovery rate (FDR) adjustment to account for multiple testing, with an FDR-adjusted *p* value of less than 0.1 considered statistically significant. To explore co-mutation patterns across *KRAS* mutation subtypes, an oncoplot and heatmap were generated using the MAFtools and ComplexHeatmap packages in R. OS was defined as the time from diagnosis of advanced disease to death in the entire cohort, and from the initiation of first-line palliative chemotherapy to death in the treated cohort. PFS was defined as the time from initiation of first-line palliative chemotherapy to documented disease progression or death from any cause. Kaplan–Meier estimates were used to evaluate OS and PFS, with subgroup comparisons performed using the log-rank test. Unstratified Cox proportional hazards regression was used to estimate hazard ratios (HRs) and their corresponding 95% confidence intervals (CIs). ORR and DCR were compared across subgroups using Fisher’s exact test. All tests were two-sided, and statistical significance was identified by a *p*-values < 0.05. Statistical analyses and data visualization were performed using GraphPad Prism version 10.3 (GraphPad Software, San Diego, CA, USA) and RStudio (version 2024, RStudio, PBC, Boston, MA) with R version 4.4.3.

## 5. Conclusions

In conclusion, while *KRAS* mutation subtypes were not significantly associated with survival outcomes in the overall cohort, our findings suggest that certain alleles, such as G12V, may be linked to differential sensitivity to standard chemotherapy. Although *KRAS* subtyping may not be universally essential for therapeutic decision-making in advanced PDAC, it could still hold clinical relevance in selected patients. Further prospective studies and functional investigations are warranted to validate these observations and clarify their implications for precision oncology.

## Figures and Tables

**Figure 1 ijms-26-10908-f001:**
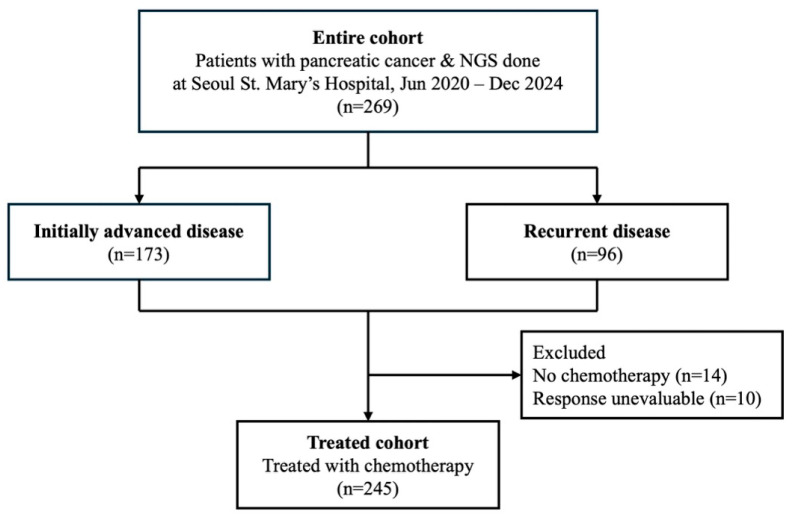
Study flowchart diagram. The flowchart shows cohort patient selection.

**Figure 2 ijms-26-10908-f002:**
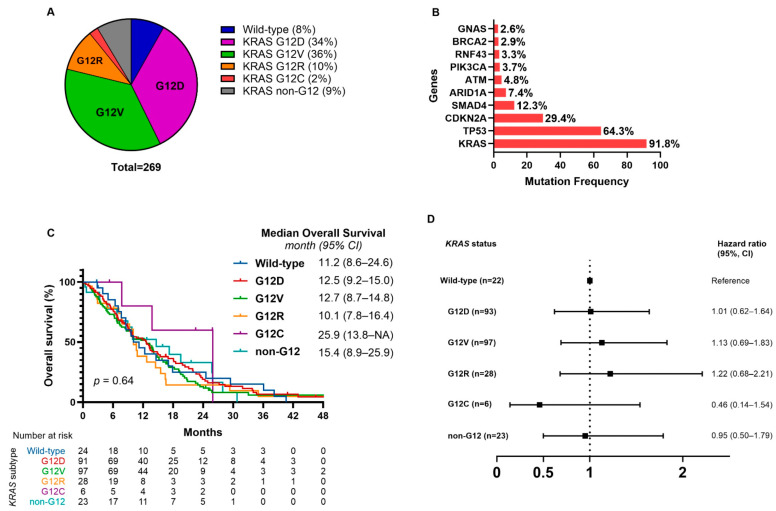
Distribution of *KRAS* mutation subtypes, frequently mutated genes, and associated overall survival in pancreatic cancer. (**A**) Distribution of *KRAS* mutation subtypes among patients with pancreatic cancer. (**B**) Bar plot illustrating the frequency distribution of the most prevalent gene mutations in our cohort. (**C**) Kaplan–Meier curves for overall survival (OS) according to *KRAS* mutation subtype. (**D**) Comparison of median OS between wild-type *KRAS* and specific *KRAS* mutation subtypes. (**A**) shows mutation-level distribution; co-occurring *KRAS* alterations within a single patient are counted separately. (**C**,**D**) are patient-level analyses, with each patient assigned to one codon group to avoid double-counting. *KRAS* G12D/V/R/C denote *KRAS* p.G12D/V/R/C. Variants follow HGVS at first mention and are abbreviated thereafter.

**Figure 3 ijms-26-10908-f003:**
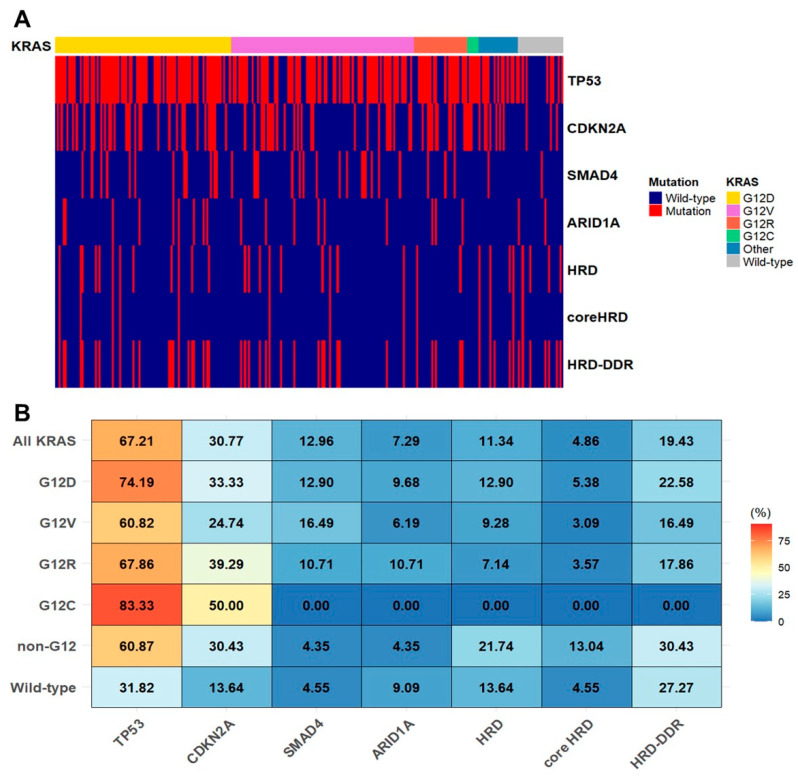
Co-occurring genomic alterations by allele specific *KRAS* mutations in pancreatic cancer. (**A**) An oncoplot illustrating somatic gene alterations stratified by *KRAS* mutation subtypes in patients with pancreatic cancer. (**B**) A heatmap showing the frequency of co-occurring genomic alterations involving *TP53*, *CDKN2A*, *SMAD4*, *ARID1A*, and genes related to homologous recombination deficiency (HRD) or DNA damage repair (DDR) across *KRAS* mutation subgroups.

**Figure 4 ijms-26-10908-f004:**
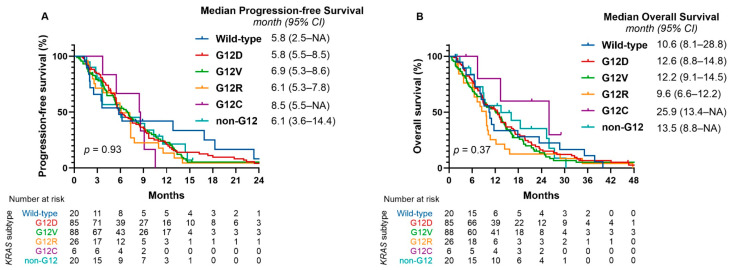
Kaplan–Meier curves for progression-free survival and overall survival by *KRAS* mutation subtype in the treated cohort. (**A**) Progression-free survival (PFS) and (**B**) overall survival (OS) according to *KRAS* mutation subtype in patients who received systemic treatment. No statistically significant differences in PFS (*p* = 0.93) or OS (*p* = 0.37) were observed among the subgroups.

**Figure 5 ijms-26-10908-f005:**
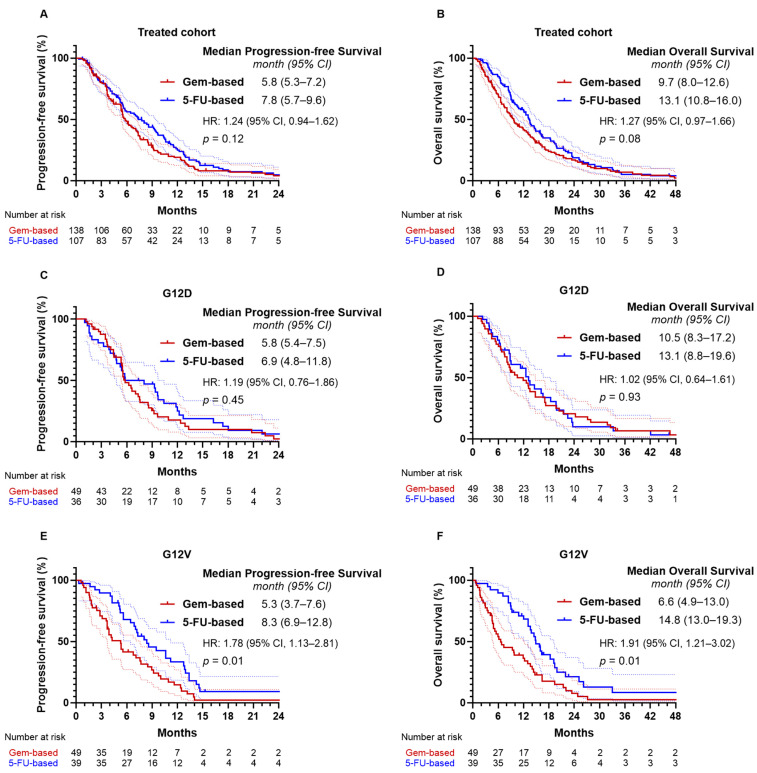
Progression-free and overall survival by first-line chemotherapy regimen in the treated cohort and *KRAS* G12D/G12V subgroups. Progression-free survival and overall survival according to first-line chemotherapy regimen in (**A**,**B**) the overall treated cohort, (**C**,**D**) the *KRAS* G12D subgroup, and (**E**,**F**) the *KRAS* G12V subgroup.

**Table 1 ijms-26-10908-t001:** Baseline characteristics of entire cohort.

Variable	Entire Cohort (n = 269)
Age, years	66 (61–73)
Gender	
Male	141 (52.4)
Female	128 (47.6)
Stage at diagnosis	
Resectable	96 (35.7)
Advanced	173 (64.3)
Disease stage	
Locally advanced	48 (17.8)
Metastatic	221 (82.2)
Tumor location	
Head	135 (50.2)
Body/Tail	134 (49.8)
Histology	
Adenocarcinoma	240 (89.2)
Others *	29 (10.8)
Grade of differentiation	
Well to moderately	153 (56.9)
Poorly	53 (19.7)
Not specified	63 (23.4)
Specimen type for NGS	
Primary tumor	176 (65.4)
Metastatic sites	76 (28.3)
Liquid biopsy	17 (6.3)
Method of specimen acquisition	
Surgical resection	115 (42.8)
Needle biopsy	137 (50.9)
Peripheral blood	17 (6.3)
*KRAS* status	
Wild type	22 (8.2)
Mutant	247 (91.8)
*TP53* status	
Wild type	96 (35.7)
Mutant	173 (64.3)
*CDKN2A* status	
Wild type	190 (70.6)
Mutant	79 (29.4)
*SMAD4* status	
Wild type	236 (87.7)
Mutant	33 (12.3)

*NGS* next-generation sequencing. Data are n (%) or median (IQR). * Other included adenosquamous, intraductal papillary mucinous neoplasm with invasive carcinoma, colloid carcinoma, acinar cell carcinoma and not specified.

**Table 2 ijms-26-10908-t002:** Baseline characteristics of treated cohort.

Variable	Entire Cohort (n = 245)
Age, years	66 (61–73)
<65	99 (40.4)
≥65	146 (59.6)
Gender	
Male	126 (51.4)
Female	119 (48.6)
ECOG performance status	
0–1	214 (87.4)
2	31 (12.6)
Disease stage	
Locally advanced	43 (17.5)
Metastatic	202 (82.5)
Previous tumor resection	
No (Initially advanced)	158 (64.5)
Yes (Recurrent disease)	87 (35.5)
Grade of differentiation	
Well to moderately	140 (57.1)
Poorly	48 (19.6)
Not specified	57 (23.3)
Number of metastatic organ sites	
0–1	170 (69.4)
≥2	75 (30.6)
Site of metastatic disease	
Liver	131 (53.4)
Lung	40 (16.3)
Peritoneum	61 (24.9)
Baseline CA 19-9 level *	
<59 × ULN (U/mL)	63 (25.7)
≥59 × ULN (U/mL)	176 (71.8)
Unknown	6 (2.5)
First-line palliative chemotherapy	
Gemcitabine-based	138 (56.3)
Gemcitabine/nab-paclitaxel	132 (53.8)
Gemcitabine	6 (2.5)
Fluorouracil-based	107 (43.7)
FOLFIRINOX	95 (38.7)
Nal-IRI/FL	12 (5.0)
*KRAS* mutation status	
Wild type	20 (8.2)
G12D	85 (34.7)
G12V	88 (35.9)
G12R	26 (10.5)
G12C	6 (2.5)
Non-G12 †	20 (8.2)

Data are n (%) or median (IQR). ECOG, Eastern Cooperative Oncology Group; CA 19-9, carbohydrate antigen 19-9; ULN, the upper limit of the normal range; FOLFIRINOX, 5-FU, leucovorin, irinotecan, and oxaliplatin; Nal-IRI/FL, fluorouracil, leucovorin, and liposomal irinotecan. * The normal range is 0–35 U/mL. CA 19-9 was stratified at 59× upper limit of normal range, consistent with prior landmark clinical trials. † Two patients harbored co-occurring *KRAS* G12D and non-G12 mutations.

**Table 3 ijms-26-10908-t003:** Treatment outcomes according to *KRAS* mutation subtypes.

Variable	Total (n = 245)	Wild Type (n = 20)	G12D (n = 85)	G12V (n = 88)	G12R (n = 26)	Non-G12 (n = 20)
Best overall response, n (%)						
Partial response	87 (35.5)	4 (20.0)	33 (38.8)	30 (34.1)	7 (26.9)	11 (55.0)
Stable disease	122 (49.8)	11 (55.0)	42 (49.4)	45 (51.1)	12 (46.2)	8 (40.0)
Progressive disease	36 (14.7)	5 (25.0)	10 (11.8)	13 (14.8)	7 (26.9)	1 (5.0)
Objective response rate, n (%)	87 (35.5)	4 (20.0)	33 (38.8)	30 (34.1)	7 (26.9)	11 (55.0)
*p* value vs. G12D		0.128	NA	0.532	0.352	0.214
Disease control rate, n (%)	209 (85.3)	15 (75.0)	75 (88.2)	75 (85.2)	19 (73.1)	19 (95.0)
*p* value vs. G12D		0.156	NA	0.656	0.115	0.686
Median PFS, months [95% CI]	6.3 [5.5–7.3]	5.8 [2.5–NA]	5.8 [5.5–8.5]	6.9 [5.3–8.6]	6.1 [5.3–7.8]	6.1 [3.6–14.4]
*p* value vs. G12D		0.718	NA	0.838	0.400	0.836
6-month PFS, % [95% CI]	51.8 [45.7–58.6]	47.9 [29.1–78.6]	49.1 [39.4–61.3]	53.5 [43.7–65.4]	53.7 [36.9–78.3]	51.1 [32.5–80.2]
Median OS, months [95% CI]	11.2 [9.6–13.1]	10.6 [8.1–28.8]	12.6 [8.8–14.8]	12.2 [9.1–14.5]	9.6 [6.6–12.2]	13.5 [8.8–NA]
*p* value vs. G12D		0.957	NA	0.496	0.146	0.714
12-month OS, % [95%]	48.0 [41.9–54.8]	33.4 [17.4–64.2]	51.8 [42.1–63.8]	50.0 [40.4–61.8]	25.4 [12.8–50.6]	55.9 [37.1–84.2]

NA not applicable; PFS progression-free survival; OS overall survival.

## Data Availability

The majority of the data generated or analyzed during this study are included in this article and its Appendix A Files. Additional datasets used in the current study are available from the corresponding author on request.

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
