# Peer review of "Clinical and Molecular Characteristics of KRAS Codon-Specific Mutations in Advanced Pancreatic Ductal Adenocarcinoma with Prognostic and Therapeutic Implications"

_ijms, 2025, doi:10.3390/ijms262210908_

Round 1

Reviewer 1 Report

Comments and Suggestions for Authors

This study has certain clinical relevance. However, the Introduction section does not clearly elaborate on the specific focus of this research and fails to effectively highlight its clinical significance. KRAS and TP53 are well-known as the most common mutations in pancreatic cancer. The major value of this study lies in the classification based on KRAS G12 mutations and its potential implications for prognosis and therapeutic guidance. It is recommended that the authors compare their findings with previous similar studies.

The Results section lacks logical clarity, making it difficult for readers to follow the authors’ intended message without frequently referring back and forth through the text. I suggest that the authors reorganize and refine the presentation of the results. In addition, I strongly suggested, it would be valuable if the authors could provide an external validation cohort to demonstrate the generalizability of the findings, rather than presenting results limited to a single center.

Author Response

Reviewer 1

Comment 1: This study has certain clinical relevance. However, the Introduction section does not clearly elaborate on the specific focus of this research and fails to effectively highlight its clinical significance. KRAS and TP53 are well-known as the most common mutations in pancreatic cancer. The major value of this study lies in the classification based on KRAS G12 mutations and its potential implications for prognosis and therapeutic guidance. It is recommended that the authors compare their findings with previous similar studies.

Response 1: Thank you for your supportive and constructive comments. We appreciate the reviewer’s suggestion to sharpen the focus and clinical significance of our work. Prior studies frequently combined patients who received curative-intent treatment with those who had advanced disease and primarily analyzed overall survival; consequently, codon-specific prognosis was inferred from heterogeneous populations without evaluating associations with systemic treatment effects. In contrast, our study restricted the cohort to advanced PDAC and, among patients who received systemic therapy, assessed not only PFS and OS but also associations with clinicopathologic features and additional measures of treatment effectiveness, including objective response rate. We believe this approach provides a more comprehensive evaluation of codon-specific prognosis in advanced PDAC. Additionally, in the opening paragraph of the Discussion, we emphasized how our study differs from prior work and highlighted our thorough evaluation of KRAS codon–specific prognosis in real-world clinical practice.

The manuscript is revised as follows:

Differences in KRAS allele-specific mutations may influence distinct signaling pathways and lead to variations in gene expression during PDAC carcinogenesis. Although each KRAS allele subtype exhibits unique biochemical and clinicopathological features, differences among KRAS allele subtypes in PDAC have not been well studied [16, 17]. While evidence on the prognostic impact of KRAS-mutated PDAC subtypes remains limited, recent studies suggest that the KRAS G12 subtype is associated with poorer outcomes than KRAS non-G12 variants [18], with KRAS G12D linked to a worse prognosis than the G12R subtype [10]. However, prior analyses combined heterogeneous disease stages and did not assess relationships with systemic treatment response, leaving the clinical relevance of KRAS subtypes uncertain. To properly evaluate co-don-specific prognosis, studies in advanced PDAC should determine whether KRAS mutation subtypes are associated with survival outcomes and the relative effectiveness of systemic chemotherapy.

In this study, we characterized the prevalence of KRAS mutation subtypes and their co-mutations in advanced PDAC and evaluated their associations with clinicopathologic features and prognostic outcomes. Additionally, we further investigated the im-pact of KRAS mutation subtypes on systemic chemotherapy effectiveness and survival outcomes.

Comment 2: The Results section lacks logical clarity, making it difficult for readers to follow the authors’ intended message without frequently referring back and forth through the text. I suggest that the authors reorganize and refine the presentation of the results.

Response 2: Thank you for this thoughtful comment. we agree with the reviewer’s aim of enhancing the logical clarity of the Results. In response, we carefully reviewed the sequencing of the Results section to ensure that the presentation follows a coherent progression and that each subsection naturally leads to the next. We also reduced unnecessary repetition and avoided excessive listing of numbers in the text, directing readers instead to the corresponding figures and tables for detailed values. In addition, we adjusted cross-references so that figures and tables align more closely with the order of presentation, improving readability and navigation.

We believe these focused edits improve the flow and clarity without altering the scientific content. If there are specific paragraphs, transitions, or data presentations the reviewer would like us to refine further, we would be grateful for more granular guidance and will readily revise accordingly. Thank you again for this constructive feedback.

Comment 3: In addition, I strongly suggested, it would be valuable if the authors could provide an external validation cohort to demonstrate the generalizability of the findings, rather than presenting results limited to a single center.

Response 3: We sincerely appreciate the reviewer’s insightful suggestion regarding an external, multicenter validation cohort. We fully agree that validation across independent institutions would strengthen generalizability and clinical applicability. Unfortunately, at the present time we are unable to assemble a multicenter cohort with harmonized treatment and survival data due to feasibility constraints. We have therefore framed our findings as hypothesis-generating and explicitly highlighted the single-center design as a key limitation. As a next step, we plan to collaborate with partner institutions to establish a multicenter dataset and prospectively assess whether the codon-stratified treatment signals observed here are reproducible.

The manuscript is revised as follows:

Our study has several limitations. First, its retrospective design and single-institution setting may introduce ascertainment bias and limit the generalizability of the findings. Second, the sample size of certain KRAS subgroups, such as G12C and non-G12 mutations, was small, reducing the statistical power to detect meaningful differences. Taken together, these two issues render the findings hypothesis-generating and warrant confirmation in adequately powered, prospective multicenter cohorts. Third, potential confounders such as performance status, disease burden, treatment dose intensity, and subsequent therapies may have influenced treatment outcomes, but multivariable analysis could not be performed owing to the limited sample size. Finally, tumor heterogeneity and temporal evolution of mutational profiles may not have been fully captured by single-sample NGS, and in a subset of patients, liquid biopsy was used, which may further affect consistency of genomic profiling.

Reviewer 2 Report

Comments and Suggestions for Authors

This study investigates the clinical and molecular characteristics of codon-specific KRAS mutations in PDAC. The scientists retrospectively evaluated 269 patients treated between 2020 and 2024 and found KRAS mutations in around 92% of cases, with p.G12V and p.G12D being the most prevalent subtypes.  Although there were no statistically significant differences in overall survival between mutation subtypes, the p.G12V subgroup appeared to benefit from fluorouracil-based treatment over gemcitabine regimens.  Co-mutation analysis revealed TP53 as a common co-alteration, particularly in G12D cancers.  The overall conclusion is that routine KRAS subtyping has minimal predictive utility but may have therapeutic applications in certain cases, particularly for the p. G12V subtype. 

Although the article is intriguing, it needs to be improved.

1) In the text illustrating Figure 2A the authors mention two co-occurring cases, creating ambiguity about whether patient counts were duplicated. The authors need to clarify their patient counting method and provide a clear table with unique patient counts by subtype, explaining and justifying any overlapping cases to ensure accurate analyses.

2) Table 2 does not clearly specify the reason for using that reference concentration for the CA19-9 biomarker, which is essential since CA19-9 stratification has been used in the analyses. 

3)  The manuscript lacks detailed methodology on NGS platforms and handling of discordant tissue/plasma results. It does not specify platform-specific limits of detection (LOD), allele frequency cutoffs for variant calling, or how tissue and plasma results were integrated into a single KRAS status. These details are crucial for variant classification and defining “KRAS wild-type.” The authors should include a clear description of LOD and allele frequency thresholds for each assay and explain how discordant results were resolved or prioritized.

  4) Survival analyses only report medians and p-values without hazard ratios or confidence intervals. Because median survival alone can be sensitive to censoring patterns, to improve accuracy and interpretation, the manuscript should include Cox hazard ratios with 95% confidence intervals for key comparisons, including treatment effects in the overall cohort and G12V subgroup, as well as subtype versus wild-type comparisons. 5)  The manuscript occasionally discusses numeric trends in very small subgroups, such as G12C (n = 6) and some non-G12 groups, and implies clinical meaning where statistical power is clearly limited. Such observations should be explicitly framed as exploratory and hypothesis-generating; conclusions based on these tiny samples must be toned down and presented with appropriate caution.

Minor

1) Figure legends for Figure 5 should include the number at risk per group .Making figure legends self-contained will prevent confusion when readers view figures separate from the text.

2) Standardize the presentation of empty or “not detected” cells between main text and supplementary data.

3) Write the protein change accurately (e.g., p.G12C) and report all gene names in italics.

4) is not clear if any patent should be described in paragraph 6. 

Author Response

Reviewer 2

This study investigates the clinical and molecular characteristics of codon-specific KRAS mutations in PDAC. The scientists retrospectively evaluated 269 patients treated between 2020 and 2024 and found KRAS mutations in around 92% of cases, with p.G12V and p.G12D being the most prevalent subtypes. Although there were no statistically significant differences in overall survival between mutation subtypes, the p.G12V subgroup appeared to benefit from fluorouracil-based treatment over gemcitabine regimens. Co-mutation analysis revealed TP53 as a common co-alteration, particularly in G12D cancers. The overall conclusion is that routine KRAS subtyping has minimal predictive utility but may have therapeutic applications in certain cases, particularly for the p. G12V subtype. Although the article is intriguing, it needs to be improved.

Comment 1: In the text illustrating Figure 2A the authors mention two co-occurring cases, creating ambiguity about whether patient counts were duplicated. The authors need to clarify their patient counting method and provide a clear table with unique patient counts by subtype, explaining and justifying any overlapping cases to ensure accurate analyses.

Response 1: We appreciate the reviewer’s careful reading. Two patients had co-occurring KRAS alterations (G12D with a non-G12 mutation). For prevalence (Figure 2A), we reported mutation-level counts, so both alterations were tallied to reflect codon occurrence. For all patient-level analyses (survival and clinicopathologic comparisons), each patient was assigned one codon using a pre-specified highest-VAF rule to represent the major subclone and avoid double-counting; in both co-occurring cases, G12D had the highest VAF and defined the codon group. We have clarified this counting strategy in the Methods and added a Figure 2 footnote indicating that Figure 2A is mutation-level (co-occurring alterations counted separately), whereas Figure 2C–2D are patient-level.

The manuscript is revised as follows:

Figure 2. Distribution of KRAS mutation subtypes, frequently mutated genes, and associated overall survival in pancreatic cancer. (A) Distribution of KRAS mutation subtypes among patients with pancreatic cancer. (B) Bar plot illustrating the frequency distribution of the most prevalent gene mutations in our cohort. (C) Kaplan–Meier curves for overall survival (OS) according to KRAS mutation subtype. (D) Comparison of median OS between wild-type KRAS and specific KRAS mutation subtypes. Figure 2A shows mutation-level distribution; co-occurring KRAS alterations within a single patient are counted separately. Figure 2C–2D are patient-level analyses, with each patient assigned to one codon group to avoid double-counting.

Molecular profiling and KRAS status were determined through tumor tissue analysis using the Oncomine Comprehensive Assay Plus or liquid biopsy via Al-phaLiquid100. These panels included 517 and 118 genes, respectively, covering onco-genes, tumor suppressor genes, gene rearrangements, microsatellite stability status, and tumor mutational burden. Formalin-fixed, paraffin-embedded (FFPE) tumor specimens or plasma samples were collected for NGS either at the time of diagnosis or upon disease recurrence. For patient-level analyses (survival and clinicopathologic comparisons), if more than one KRAS alteration was detected in a single patient, we assigned a single codon category based on the highest variant allele frequency to avoid double-counting.

Comment 2: Table 2 does not clearly specify the reason for using that reference concentration for the CA19-9 biomarker, which is essential since CA19-9 stratification has been used in the analyses.

Response 2: We appreciate the reviewer’s insightful comment. While there is no universally accepted prognostic cut-off for CA19-9 in pancreatic cancer, prior studies have associated very high baseline CA19-9 with poorer outcomes—beginning with an early report that used ≥59× the upper limit of normal (ULN) as an adverse threshold (Lancet Oncol. 2008;9:132–138). Subsequent landmark phase III trials of FOLFIRINOX and gemcitabine plus nab-paclitaxel also summarized baseline characteristics and exploratory analyses using the same 59×ULN categorization (N Engl J Med 2011;364:1817–1825; N Engl J Med 2013;369:1691–1703). For clinical interpretability and comparability with these references, we adopted 59×ULN for stratification in our analyses. We have revised Table 2 to add a footnote stating that CA19-9 was stratified at 59× ULN, consistent with prior landmark clinical trials.

The manuscript is revised as follows:

Serum tumor markers, including carbohydrate antigen 19-9 (CA 19-9) and carcinoembryonic antigen (CEA), were measured concurrently with radiologic assessments. For stratified analyses, we applied a CA 19-9 threshold of 59× the upper limit of normal consistent with landmark clinical trials [4, 5].

*The normal range is 0–35 U/mL. CA 19-9 was stratified at 59×upper limit of normal range, consistent with prior landmark clinical trials. †Two patients harbored co-occurring KRAS G12D and non-G12 mutations.

Comment 3: The manuscript lacks detailed methodology on NGS platforms and handling of discordant tissue/plasma results. It does not specify platform-specific limits of detection (LOD), allele frequency cutoffs for variant calling, or how tissue and plasma results were integrated into a single KRAS status. These details are crucial for variant classification and defining “KRAS wild-type.” The authors should include a clear description of LOD and allele frequency thresholds for each assay and explain how discordant results were resolved or prioritized.

Response 3: We appreciate the reviewer’s perceptive comments. We agree that a reliable definition of KRAS wild-type requires transparent reporting of assay performance, including platform-specific LODs and allele-frequency thresholds. Because explicit LODs are not uniformly reported across platforms, we have provided the prespecified variant-calling cutoffs used in our analyses. In our cohort, plasma NGS was performed only when tissue NGS was unavailable; therefore, direct tissue–plasma discordance could not be assessed. We have revised the Methods to specify the allele-frequency thresholds for the tissue (Oncomine Comprehensive Assay Plus) and plasma (AlphaLiquid100) assays.

The manuscript is revised as follows:

Genomic DNA and RNA were extracted from FFPE tumor samples and analyzed using the Oncomine Comprehensive Assay Plus (Thermo Fisher Scientific, Waltham, MA, USA) on the IonTorrent™ S5 XL platform, following the manufacturer’s protocols. Sequencing quality was assessed using plug-in coverage analysis, and data were processed through dedicated bioinformatic workflows within the Ion Reporter v5.12 server (IonTorrent, Waltham, MA, USA). Variants, including single nucleotide variants (SNVs), insertions/deletions (indels), copy number variations (CNVs), and gene fusions, were filtered based on the following thresholds: a minimum allele frequency of ≥ 1.5% for hotspot SNVs and indels, a minimum read count of 100 for fusions, and CNV thresholds of ≥ 6 for gains and ≤ 0.7 for losses. Variants were classified into clinical relevance tiers according to established guidelines [19].

Cell-free DNA (cfDNA) was extracted from plasma samples prior to systemic treatment. DNA NGS library preparation and solution-based target enrichment were performed at IMBdx, Inc. (Seoul, Korea) using AlphaLiquid100, as previously described [20]. A targeted panel of 118 cancer-related genes was used to detect four variant classes (SNVs, indels, CNVs, and fusions). Captured DNA libraries were sequenced on the Il-lumina NovaSeq 6000 platform (Illumina, San Diego, CA, USA) in 2×150 bp paired-end mode. For AlphaLiquid100, variants were called using thresholds of at least 0.1% allele fraction for SNVs and indels (high-confidence at 0.5% or higher), 2.4 copies or higher for CNVs, and 0.2% or higher for fusions.

Comment 4: Survival analyses only report medians and p-values without hazard ratios or confidence intervals. Because median survival alone can be sensitive to censoring patterns, to improve accuracy and interpretation, the manuscript should include Cox hazard ratios with 95% confidence intervals for key comparisons, including treatment effects in the overall cohort and G12V subgroup, as well as subtype versus wild-type comparisons.

Response 4: Thank you for the reviewer’s insightful comment. In response, we now present the median overall survival for the entire cohort in Figure 2C, and the comparisons between KRAS wild-type and individual KRAS subtypes using hazard ratios with 95% confidence intervals in Figure 2D. For the treated cohort, we have added hazard ratios with 95% CIs for both PFS and OS in Figure S1. We also updated Figure 5 to include hazard ratios with 95% CIs for PFS and OS stratified by treatment regimen. We believe these additions enhance clarity and address the reviewer’s concern.

The manuscript is revised as follows:

In the overall treated cohort, the median progression-free survival (PFS) was 6.3 months (95% CI, 5.5–7.3), and the median OS was 11.2 months (95% CI, 9.6–13.1). Sur-vival analysis revealed no statistically significant difference in PFS among the KRAS mutation subtypes (p = 0.93; Figure 4A). Similarly, no significant difference in OS was observed between the subgroups (p = 0.37; Figure 4B). Findings from very small sub-groups are exploratory and should be interpreted with caution. Compared with KRAS wild-type, none of the codon-specific KRAS subtypes exhibited statistically significant differences in PFS or OS (Figure S1).

Figure S1. Comparison of median progression-free survival (A) and overall survival (B) between KRAS wild-type and KRAS codon-specific subtypes in the treated cohort.

Comment 5: The manuscript occasionally discusses numeric trends in very small subgroups, such as G12C (n = 6) and some non-G12 groups, and implies clinical meaning where statistical power is clearly limited. Such observations should be explicitly framed as exploratory and hypothesis-generating; conclusions based on these tiny samples must be toned down and presented with appropriate caution.

Response 5: Thank you for the reviewer’s insightful comment. We agree that results from very small subgroups (such as KRAS G12C and certain non-G12 groups) are underpowered. Accordingly, we have removed language in the Results section that described a numerically longer OS in the absence of statistical significance. We also revised the Limitations to explicitly note the very small sample sizes in these subgroups. Finally, while the G12C subgroup represents a druggable alteration and some patients (not included in the present analysis) may derive survival benefit from targeted therapy, we have confined this point to a brief, contextual note in the Discussion.

The manuscript is revised as follows:

In the overall treated cohort, the median progression-free survival (PFS) was 6.3 months (95% CI, 5.5–7.3), and the median OS was 11.2 months (95% CI, 9.6–13.1). Sur-vival analysis revealed no statistically significant difference in PFS among the KRAS mutation subtypes (p = 0.93; Figure 4A). Similarly, no significant difference in OS was observed between the subgroups (p = 0.37; Figure 4B). Findings from very small sub-groups are exploratory and should be interpreted with caution. Compared with KRAS wild-type, none of the codon-specific KRAS subtypes exhibited statistically significant differences in PFS or OS (Figure S1).

Comment 6: Figure legends for Figure 5 should include the number at risk per group .Making figure legends self-contained will prevent confusion when readers view figures separate from the text.

Response 6: We appreciate the reviewer’s careful comment. We have revised Figure 5 to include the numbers at risk for each group.

The manuscript is revised as follows:

Comment 7: Standardize the presentation of empty or “not detected” cells between main text and supplementary data.

Response 7: Thank you for the insightful comment. In the main text, cells that are empty because the item is not applicable are now denoted as “NA.” In Supplementary Table S1, cases in which a mutation was not detected are labeled “not detected.” We also reviewed the manuscript and all supplementary materials to ensure consistent usage and corrected any remaining inconsistencies.

Comment 8: Write the protein change accurately (e.g., p.G12C) and report all gene names in italics.

Response 8: We appreciate the reviewer’s insightful comment. We have revised the manuscript to standardize variant nomenclature. Specifically, we now follow Human Genome Variation Society (HGVS) guidelines at first mention (e.g., KRAS p.G12D) and use the abbreviated form thereafter (e.g., G12D). Gene symbols are italicized throughout. We also added a brief note clarifying this convention at the first occurrence in the Introduction (and referenced it in figure legends as needed) to ensure consistency and clarity across the text and figures.

The manuscript is revised as follows:

The majority of KRAS missense mutations occur at codon 12, with p.G12D being the most common (35–40%), followed by p.G12V (20–30%), p.G12R (10–20%), and other rare variants [9, 10]. (subsequently abbreviated as G12D, G12V, and G12R).

Figure 2. Distribution of KRAS mutation subtypes, frequently mutated genes, and associated overall survival in pancreatic cancer. (A) Distribution of KRAS mutation subtypes among patients with pancreatic cancer. (B) Bar plot illustrating the frequency distribution of the most prevalent gene mutations in our cohort. (C) Kaplan–Meier curves for overall survival (OS) according to KRAS mutation subtype. (D) Comparison of median OS between wild-type KRAS and specific KRAS mutation subtypes. Figure 2A shows mutation-level distribution; co-occurring KRAS alterations within a single patient are counted separately. Figure 2C–2D are patient-level analyses, with each patient assigned to one codon group to avoid double-counting. KRAS G12D/V/R/C denote KRAS p.G12D/V/R/C. Variants follow HGVS at first mention and are abbreviated there-after.

Comment 9: is not clear if any patent should be described in paragraph 6.

Response 9: Thank you for raising this point. We confirm that there are no patents related to this work.

Round 2

Reviewer 1 Report

Comments and Suggestions for Authors

I have no further comments.

Reviewer 2 Report

Comments and Suggestions for Authors

The manuscript can be accepted without any further changes.